# Effects of Inosine-5′-monophosphate Dehydrogenase (IMPDH/GuaB) Inhibitors on *Borrelia burgdorferi* Growth in Standard and Modified Culture Conditions

**DOI:** 10.3390/microorganisms12102064

**Published:** 2024-10-15

**Authors:** Eric L. Siegel, Connor Rich, Sanchana Saravanan, Patrick Pearson, Guang Xu, Stephen M. Rich

**Affiliations:** 1Laboratory of Medical Zoology, Department of Microbiology, University of Massachusetts, Amherst, MA 01003, USA; esiegel@umass.edu (E.L.S.); crich@umass.edu (C.R.); sanchanasara@umass.edu (S.S.); pbpearson@umass.edu (P.P.); gxu@umass.edu (G.X.); 2New England Center of Excellence in Vector-Borne Disease, Amherst, MA 01003, USA

**Keywords:** *Borrelia burgdorferi*, inosine-5′-monophosphate dehydrogenase, GuaB, Lyme disease

## Abstract

*Borrelia burgdorferi*’s inosine-5′-monophosphate dehydrogenase (IMPDH, GuaB encoded by the *guaB* gene) is a potential therapeutic target. GuaB is necessary for *B. burgdorferi* replication in mammalian hosts but not in standard laboratory culture conditions. Therefore, we cannot test novel GuaB inhibitors against *B. burgdorferi* without utilizing mammalian infection models. This study aimed to evaluate modifications to a standard growth medium that may mimic mammalian conditions and induce the requirement of GuaB usage for replication. The effects of two GuaB inhibitors (mycophenolic acid, 6-chloropurine riboside at 125 μM and 250 μM) were assessed against *B. burgdorferi* (*guaB*+) grown in standard Barbour–Stoenner–Kelly-II (BSK-II) medium (6% rabbit serum) and BSK-II modified to 60% concentration rabbit serum (BSK-II/60% serum). BSK-II directly supplemented with adenine, hypoxanthine, and nicotinamide (75 μM each, BSK-II/AHN) was also considered as a comparison group. In standard BSK-II, neither mycophenolic acid nor 6-chloropurine riboside affected *B. burgdorferi* growth. Based on an ANOVA, a dose-dependent increase in drug effects was observed in the modified growth conditions (F = 4.471, *p* = 0.001). Considering higher drug concentrations at exponential growth, mycophenolic acid at 250 μM reduced spirochete replication by 48% in BSK-II/60% serum and by 50% in BSK-II/AHN (*p* < 0.001 each). 6-chloropurine riboside was more effective in both mediums than mycophenolic acid, reducing replication by 64% in BSK-II/60% serum and 65% in BSK-II/AHN (*p* < 0.001 each). These results demonstrate that modifying BSK-II medium with physiologically relevant levels of mammalian serum supports replication and induces the effects of GuaB inhibitors. This represents the first use of GuaB inhibitors against *Borrelia burgdorferi*, building on tests against purified *B. burgdorferi* GuaB. The strong effects of 6-chloropurine riboside indicate that *B. burgdorferi* can salvage and phosphorylate these purine derivative analogs. Therefore, this type of molecule may be considered for future drug development. Optimization of this culture system will allow for better assessment of novel Borrelia-specific GuaB inhibitor molecules for Lyme disease interventions. The use of GuaB inhibitors as broadcast sprays or feed baits should also be evaluated to reduce spirochete load in competent reservoir hosts.

## 1. Introduction

Lyme disease is the most prevalent vector-borne disease in the United States [1]. Its etiological agent, *Borrelia burgdorferi* sensu stricto, confers an estimated 476,000 annual cases through bites from the blacklegged tick (*Ixodes scapularis*) [2]. Timely antibiotic treatment usually results in complete recovery [3]. However, patients may experience prolonged clinical signs associated with secondary diseases of the cardiac, nervous, and other systems [4]. These severe manifestations may worsen when coupled with co-infections of other tick-borne pathogens [5]. New, effective therapeutics targeting human infections and the *B. burgdorferi* enzootic cycle are needed to address limitations, including possible resistance arising to available conventional treatments and incomplete effectiveness.

Nucleotide biosynthesis is a fundamental process in cellular replication, function, and homeostasis [6]. Like other obligate pathogens with compact genomes, *Borrelia* spp. lack the requisite pathways to perform de novo nucleotide biosynthesis [7]. Instead, they must rely on an essential pathway of salvage and interconversion of exogenous precursors to meet replication needs [8]. Limiting steps of similar pathways have been exploited for virulence- and proliferation-disrupting therapeutics for mammalian hosts. One example of successful application was the development of pyrimethamine, which selectively disrupts dihydrofolate reductases of *Plasmodium* spp. and *Toxoplasma gondii* with minimal toxicity to the mammalian host [9,10]. Developing drugs that target the essential enzymes for nucleotide biosynthesis requires selective targeting to minimize toxicity and off-target effects. This challenge arises because the divergence of microbial enzymes from mammalian orthologs complicates binding affinities, thus reducing effectiveness, bioavailability, and ease of delivery.

*Borrelia burgdorferi*’s inosine-5′-monophosphate dehydrogenase (IMPDH, GuaB encoded by the *guaB* gene) is a critical enzyme that catalyzes the limiting, NAD-mediated interconversion of inosine-5′-monophosphate (IMP) to xanthosine-5′-monophosphate (XMP). Xanthosine-5′-monophosphate is then converted to guanosine monophosphate (GMP) by GMP synthase for use in DNA/RNA [11]. Mycophenolic acid and ribavirin are examples of GuaB inhibitors that have proven effective for numerous immunological, antiviral, and anti-tumor therapies [11]. Differences in the catalytic behaviors and active sites of mammalian and microbial GuaB equivalents are well-defined. The binding affinities and effectiveness of available GuaB inhibitors are largely ineffective or too poorly selective for use against pathogenic microorganisms in mammalian hosts. For example, mycophenolic acid is shown to inhibit the mammalian enzyme at much lower concentrations (human 10–20 nM) than those for *B. burgdorferi* (7.9 μM) [11]. Therefore, species-specific GuaB inhibitors would be needed to target *B. burgdorferi* without causing severe harm to the host.

*Borrelia* spp. are maintained in distinct environments across their enzootic cycle [12]. These include the tick midgut and the tissues of reservoir and incidental/dead-end mammalian hosts. These environments present disparate relative abundances of purines available for scavenging and thus dictate the use of different pathways to utilize these resources in nucleotide synthesis. The *B. burgdorferi* GuaB is essential for utilizing scavenged hypoxanthine and adenine [13]. Hypoxanthine is most abundant in blood plasma available to *B. burgdorferi* while transiently residing in the bloodstream during localized infection. Adenine availability may increase with spirochete dissemination to cardiac, dermal, bladder, and other tissues [14]. Mammalian tissues are deficient in guanine, which dictates the essentiality of the *guaB* pathway [13,14]. By comparison, in the tick midgut, a fitness advantage is only seen with *guaB*+ spirochetes relative to knockout strains [13]. The survival of *guaB*-deficient spirochetes in the tick midgut has been associated with the demonstration of direct transport of guanine deoxyribonucleotides and ribonucleotides for use in DNA and RNA synthesis [15]. This direct transport mechanism would allow *B. burgdorferi* to circumvent its reliance on the *guaB* pathway when a sufficient source of free guanine is present. *guaB*-deficient spirochetes also do not exhibit a defect in vitro (when grown in nutrient-rich Barbour–Stoenner–Kelly-II (BSK-II) medium) [13]. A similar mechanism of guanine utilization has been hypothesized to explain this.

Solving the crystal structure of *B. burgdorferi’s* GuaB has made it possible to manipulate available inhibitors and develop new molecules that may selectively target *B. burgdorferi* [16]. However, these inhibitors cannot be tested in vitro due to the lack of an appropriate culture medium. An in vitro assay fills the crucial gap between testing new drugs in silico or against the purified enzyme before moving into the mammalian model. This report describes novel culture conditions that sustain *B. burgdorferi* growth and induce borreliastatic effects of GuaB inhibitors. We demonstrate effects using a molecule that has yet to be assessed against *B. burgdorferi*, 6-chloropurine riboside, and mycophenolic acid, which has been tested against purified GuaB [17]. This work represents the first instance of assessing the impacts of GuaB inhibitors against *B. burgdorferi* beyond work with the purified enzyme. We considered three growth conditions based on standard BSK-II. We modified this medium with higher concentrations of rabbit serum, hypothesizing that physiologically relevant conditions can induce effects due to a shift in the relative abundance of purine precursors for scavenging. To further gather evidence for this hypothesis, we used a third growth condition, modifying standard BSK-II with the direct supplementation of exogenous hypoxanthine and adenine, in addition to nicotinamide, the precursor to NAD needed for the GuaB-mediated reaction. This work outlines the needs and path for optimizing this system to provide the test system for new GuaB inhibitors.

## 2. Materials and Methods

Cultures were prepared from BSK-II medium formulated in-house and supplemented with rifampicin [50 μg/mL], fosfomycin [20 μg/mL], and amphotericin B [2.5 μg/mL] [18]. Cultures of roughly 300,000 spirochetes/mL starting concentrations were prepared in 1.5 mL volumes using *B. burgdorferi* strain B31, clone 5A1 (BEI Resources NR-13251, Manassas, VA, USA). Three growth conditions were considered: standard BSK-II; BSK-II with added non-hemolyzed rabbit serum (Pel-Freez Biologicals 31125-5, Rogers, AR, USA) at 60%; and a comparison group of BSK-II with added hypoxanthine (Sigma-Aldrich 68-94-0, St. Louis, MA, USA), adenine (Sigma-Aldrich 73-24-5), and nicotinamide (Sigma-Aldrich 98-92-0) at 75 μM each in Dimethyl sulfoxide (DMSO) as a vehicle (BSK-II/AHN). Technical grade mycophenolic acid (≥98%, Sigma-Aldrich 24280-93-1) and 6-chloropurine riboside (99%, Sigma-Aldrich 5399-87-1) were formulated in DMSO for testing at 125 μM and 250 μM. Controls consisted of vehicle-only (DMSO) groups. The concentration of DMSO was less than 0.5% of the culture, which was previously shown to have no harmful effects on growing spirochetes in standard cultures [19]. Solutions were added to the culture at day 0 of incubation in a water jacket incubator at 34 °C. Triplicate counts were performed using disposable hemocytometers under dark-field microscopy. Preliminary confirmation that BSK-II with higher concentrations of rabbit serum supported growth was performed and is found in the Appendix A.

To compare drug effects between growth conditions, we applied an ANOVA to normalized data, with growth conditions and treatment as fixed factors. To obtain a normalization constant for each growth condition, we calculated the ratio of growth for triplicate counts relative to the mean of respective controls. This approach accounted for observed differences in absolute growth between groups without deflating observed variance. Pairwise comparisons of treatments between groups were made with Tukey’s HSD correction, focusing on peak exponential growth. The analysis was performed in R Studio (Boston, MA, USA).

## 3. Results

Spirochete replication trends differed slightly between growth conditions (Figure 1). Higher replication rates were generally observed after day 4. Replication rates were highest in standard BSK-II. The trend lines of replication at the onset of exponential growth were consistent for DMSO controls and inhibitor groups, except for 125 μM mycophenolic acid in BSK-II/AHN. This group interestingly exhibited a spike in growth a day prior to the other groups in this condition.

No effects of mycophenolic acid or 6-chloropurine riboside at either test concentration were observed in standard BSK-II (Figure 1a). However, similar dose-dependent effects on replication were observed in both modified BSK-II formulations. In BSK-II/60% and BSK-II/AHN, both mycophenolic acid and 6-chloropurine riboside induced a flatter exponential pattern at 250 μM (Figure 1b,c). At exponential growth, *B. burgdorferi* exposed to mycophenolic acid showed a reduction in growth by 48% in BSK-II/60% and 50% in BSK-II/AHN at 250 μM. The effects were weaker for mycophenolic acid at 125 μM, showing reductions in growth by 38% in BSK-II/60%, and it was not effective in BSK-II/AHN. 6-Chloropurine riboside was ineffective at 125 μM in BSK-II/60% but showed some activity in BSK-II/AHN, resulting in a 30% reduction. 6-Chloropruine riboside at 250 μM exhibited the strongest effects on growth, showing reductions of 64% in BSK-II/60% and 65% in BSK-II/AHN.

A significant interaction between environment and treatment (F = 4.471, *p* = 0.001) was shown with an ANOVA (Table 1). Considering pairwise comparisons, the effects of 250 μM mycophenolic acid (BSK-II/60%, *p* = 0.0002; BSK-II/AHN, *p* = 0.0006) and 250 μM 6-chloropurine riboside (BSK-II/60%, *p* = 0.0007; BSK-II/AHN, *p* = 0.0001) were significantly different between standard BSK-II and BSK-II/60% and BSK-II/AHN (Table 2). 6-Chloropurine riboside at 125 μM was ineffective in BSK-II/60% but showed mild, significant effects in BSK-II/AHN (*p* = 0.0243). The effects of mycophenolic acid in BSK-II/60% and BSK-II/AHN at 125 μM were not significantly different from those in standard BSK-II (Figure 2).

## 4. Discussion

This report indicates that GuaB inhibitors can induce borreliastatic activity in culture if the base concentration of rabbit serum in BSK-II is increased. The most likely explanation for the induction of drug effects is that higher concentrations of rabbit serum will dilute the purine pool in BSK-II for spirochetes to salvage, flooding otherwise accessible free guanine sources with hypoxanthine, adenine, and other unrelated molecules. This dilution would then prompt the requisite increase in *GuaB* expression, which is likely otherwise likely expressed at low levels in BSK-II, to support replication. Therefore, this situation simulates an exploitative model of the essentiality pattern of *GuaB* observed in the mammalian environment. This hypothesis is supported by the consistency of the effects of higher concentrations of the inhibitors tested between the modified growth mediums. Increased serum is likely preferred to direct additions of hypoxanthine, adenine, and nicotinamide as it is more representative of host conditions through spirochete dissemination. The differences observed in growth trends between the two modified conditions warrant further investigation.

We modified BSK-II by increasing the concentration of rabbit serum from the standard 6% to 60%. This represents a physiologically relevant concentration to which *B. burgdorferi* is exposed when interacting with the mammalian bloodstream. Plasma (approximately 55% of blood concentration) includes mostly water, proteins, and purines, among other molecules accessible for scavenging. *Borrelia* can efficiently salvage purines at low concentrations. Spirochetes need to be able to use this supply to establish their initial presence in the bloodstream. Hypoxanthine is overwhelmingly more abundant in rabbit plasma (5.1 μM, approximately 25×) than adenine and guanine (<0.2 μM) [20]. It may be possible that similar serum concentrations in culture can mimic these conditions and induce *guaB* at levels observed with mammalian infection. While our results suggest that *guaB* expression is upregulated in this condition, it is still unknown how its expression is affected by the presence of the other nutrient-rich components in BSK-II. For example, we saw that despite being unable to sustain high levels of replication in the presence of inhibitors, the direct addition of adenine and hypoxanthine supported an initial level of growth that was not seen in the 60% serum group. If there are unidentified cues for the expression of *guaB* that may govern these differences (present in plasma, for example), these could be investigated by connecting biochemical characterization with gene expression.

Further investigation into this would require a series of experiments. In the process of conducting these, we could further optimize media to provide a system to test new drugs in a translatable environment. The first step would be to titrate BSK-II stocks with rabbit serum at a fine scale. The relative abundance of guanine/hypoxanthine/adenine in each culture must be compared to isolated mammalian blood using biochemical means. The hypothesis that components of BSK-II may be a source of excess nucleosides/nucleotides available for scavenging can also be confirmed at this time. Spirochetes can be grown in each condition, and the expression of *guaB* can be compared between conditions and with expression levels in the mammalian host using an already developed *guaB* reverse transcription-quantitative polymerase chain reaction (rt-qPCR) [13]. Once an appropriate medium formulation has been chosen, GuaB inhibitors can be added to the culture and assessed with inhibition assays. This is essential for confirming that the effects we observe are due to GuaB inhibition. Taking 6-chloropurine riboside, for example, this molecule targets the IMP binding pocket of GuaB. Antimetabolite effects at other points in the salvage, transport, and interconversion pathway may compound the results observed. This experiment was, however, an important step as it was unknown whether *B. burgdorferi* could uptake, phosphorylate, and use this analog-type molecule in its nucleotide biosynthesis pathways. This also highlights the need to test these molecules against whole spirochetes instead of jumping from enzyme inhibition tests to mammalian tests. With these experiments, we can develop conditions for spirochete growth that activate the purine salvage and interconversion pathways used in the mammalian host. Therefore, this would create an effective test system mitigating the need for murine models until optimal compounds are developed.

Other considerations also warrant investigation based on observations in this report. The peak cell concentration in controls grown in BSK-II/60% was not as high as expected based on preliminary experiments (Appendix A). This could be due to the addition of DMSO, the effects of which still need to be tested with increased serum concentrations, despite having been demonstrated to be acceptable at the concentrations used as a vehicle in standard growth media. Differences in growth could also be attributed to differences between lots of rabbit serum or sources, which may vary in exact composition. We also used *B. burgdorferi* 5A1, a laboratory strain that has well-studied properties. This strain is understood to grow well and is invasive to the range of mammalian hosts. It is of interest to study effects in naturally occurring human-invasive strains and perform more intensive biological replicates to accurately characterize effects.

The treatment for Lyme disease is stage-dependent. Doxycycline (for 10–14 days) is the most commonly prescribed antibiotic for Lyme disease without neurologic, cardiac, or joint involvement [3]. Amoxicillin, cefuroxime, and azithromycin may also be used. Treatment is more complex with manifestations associated with secondary diseases of major systems [5]. While available antibiotics are usually effective, there remains a need to have more tools available to address clinical disease. Future work is still needed to better understand the causes of the persistence of clinical signs and incomplete efficacy of treatments, such as the presence of pleomorphic forms. Novel GuaB inhibitors could add to available treatments and provide another means of Lyme disease intervention. Alternatively, the most effective use case for GuaB inhibitors could be aimed at reducing the prevalence of *B. burgdorferi* in small mammal reservoir hosts, such as the white-footed mouse (*Peromyscus leucopus*) and eastern chipmunk (*Tamias striatus*), which are most abundant in the Northeast, United States [21]. As *guaB* is expressed several weeks post-infection in murine models, inhibitors in the form of broadcast sprays or feed baits could selectively target *B. burgdorferi* and reduce the spirochete load in the reservoir host. This could result in a reduced prevalence of *B. burgdorferi*-infected hosts on which immature *Ixodes* ticks feed and the subsequent reduction in the infection prevalence of ticks that feed on these treated reservoir hosts. Doxycycline could theoretically apply to this use case; however, concern about developing resistance in the environment for our most prominently used compound prevents this from being practical. Therefore, GuaB inhibitors are one way to fill the need for compounds in this application.

Targeting *B. burgdorferi* spirochetes in the small mammal reservoir host could also impact the proliferation of other tick-borne pathogens. For example, invasive strains of *B. burgdorferi* have been shown to promote the transmission of *Babesia microti* [22]. Future studies could also consider using GuaB inhibitors to target *B. microti* and other tick-borne pathogens directly. Foundational work for this has been established with a study demonstrating the inhibition of *B. gibsoni* and *B. bovis* propagation using mycophenolic acid, mycophenolate mofetil, mizoribin, ribavirin, and 7-nitroindole [23]. The possibility of resistance to these treatments and the underlying mechanisms should also be studied before such interventions are trialed.

## 5. Conclusions

We demonstrate that BSK-II can be manipulated with higher concentrations of rabbit serum to induce suppression of spirochete growth with GuaB inhibitors, where they would otherwise be ineffective. Therefore, the effects of GuaB inhibitors are shown in culture for the first time. This is a foundation for future work that may optimize growth conditions with more physiological relevance to *guaB* expression patterns. An optimized assay will permit the assessment of species-specific GuaB inhibitors intended for mammalian administration before testing in murine/other animal models. This could provide an easy, accessible alternative to preliminary testing of these molecules.

## Figures and Tables

**Figure 1 microorganisms-12-02064-f001:**
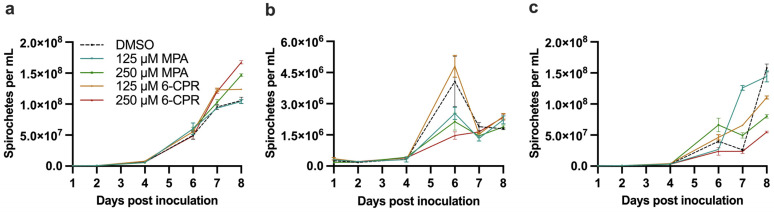
Observed growth of *Borrelia burgdorferi* 5A1 with GuaB inhibitors under three conditions: (**a**) standard BSK-II; (**b**) modified BSK-II with 60% rabbit serum; (**c**) BSK-II with direct supplementation of 75 μM adenine, hypoxanthine, and nicotinamide. Abbreviations: MPA = mycophenolic acid; 6-CPR = 6-Chloropurine riboside. DMSO 0.5% (dashed line) represents the vehicle-only control group in each panel.

**Figure 2 microorganisms-12-02064-f002:**
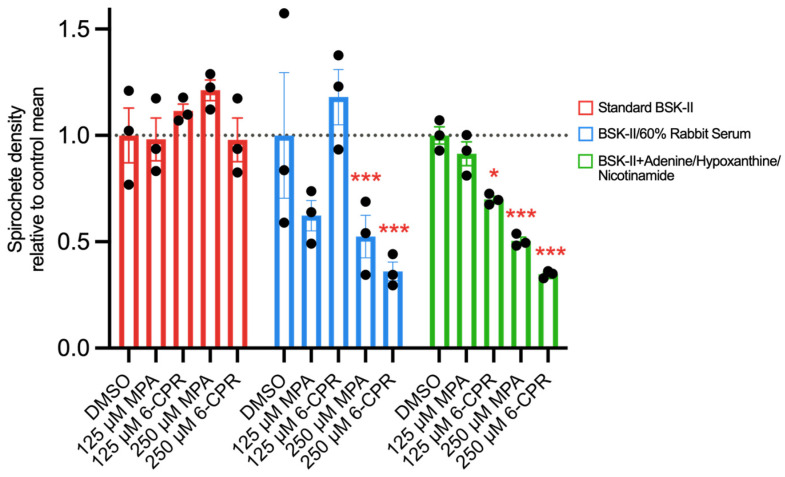
GuaB inhibitor effects on *Borrelia burgdorferi* growth under three growth conditions at exponential growth. Spirochete concentration is shown relative to the controls of the same growth condition. The significance of the interaction between treatment and growth condition is shown in Table 2. Abbreviations: MPA = mycophenolic acid; 6-CPR = 6-chloropurine riboside; DMSO = dimethyl sulfoxide (to 0.5% concentration of medium) used as the negative control. Significance (* *p* < 0.05; *** *p* < 0.001) is shown for pairwise comparisons, detailing differences in inhibitor effects between modified growth conditions and standard BSK-II (estimates and exact *p* values are shown in Table 2).

**Table 1 microorganisms-12-02064-t001:** Main effects (growth condition and treatment) and their interaction by ANOVA on spirochete replication.

Effect	df	Sum Sq	Mean Sq	F Value	Pr (>F)
Condition	2	1.186	0.5932	17.743	<0.001
Treatment	4	1.223	0.3057	9.144	<0.001
Condition × Treatment	8	1.196	0.1495	4.471	0.001
Residuals	30	1.003	0.0334		

**Table 2 microorganisms-12-02064-t002:** Comparisons for growth condition x treatment interaction with standard BSK-II.

Treatment	BSK-II/60% Rabbit Serum Estimate (*p* Sig.)	Direct Addition of Hypoxanthine, Adenine, Nicotinamide Estimate (*p* Sig.)
250 μM mycophenolic acid	0.6879 (*p* = 0.0002)	0.7072 (*p* = 0.0001)
125 μM mycophenolic acid	0.3586 (*p* = 0.0574)	0.0676 (*p* = 0.8936)
250 μM 6-chloropurine riboside	0.6186 (*p* = 0.0007)	0.6325 (*p* = 0.0006)
125 μM 6-chloropurine riboside	−0.0650 (*p* = 0.9010)	0.4155 (*p* = 0.0243)

## Data Availability

All relevant data are included in the manuscript or Appendix A.

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
