# Peer review of "Effects of Inosine-5′-monophosphate Dehydrogenase (IMPDH/GuaB) Inhibitors on Borrelia burgdorferi Growth in Standard and Modified Culture Conditions"

_microorganisms, 2024, doi:10.3390/microorganisms12102064_

Round 1
Reviewer 1 Report
Comments and Suggestions for Authors
Attached

Reviewer 2 Report
Comments and Suggestions for Authors
In the reviewed MS the authors investigated how different modifications of standard medium BSK-II, developed earlier for cultivating pathogenic Borrelia bacteria, influence effects of two inhibitors (myc.acid and 6-Chloropurine) of GuaB (a nucleotide biosynthesis enzyme) on the bacterial growth. They modified the standard medium in two ways: (1) applied 60% rabbit serum, (2) added 3 different compounds (exogenous hypoxanthine, adenine, nicotinamide) in 75 mM concentration to the standard medium. The authors showed that the inhibitors indeed works better in the modified medium. The MS is well written. Statistical methods were applied correctly. The Figures and captions need moderate revisions. The MS needs only minimal linguistic polishing. Supplementary files mentioned several times in the text, are unavailable for reviewing. The Abstract needs serious revision. It is confusing for readers in its current form. Several additional remarks are below.
11 GuaB – please, explain the origin of this abbreviation
12,13 please, reword this phrase, currently it may be unclear for readers
19 burgdorferi 5A1 – could you specify if this is guaB+ or guaB- strain?
20 In standard BSK-II – is it the medium with 60% albumin or not? Please, specify this
20,21 (no effect) contradict 23,24 (effect yes)
24 what 48 and 50 correspond to?
25 64 and 65 – same question
26-27 do you mean 60% serum modification here or not?
25 vs 28 please, compare and avoid repetition
29, 31 – check italic for Borrelia in the whole MS
73 please explain that guaB is a gene and explain why the gene name and the enzyme name are so different. This may be unclear for readers
109 B31 – please, add a brief phrase explaining this abbreviation.
116 please, mention these two concentrations in the Abstract
121 is found in the supplemental data – no supplementary files are provided
133 vs 134-136 please compare these statements, they are contradictory
137 italic
138 48%, 50% (250 μM) and 38%, 9% (125 mM) – please, explain why you give two % numbers prior to the concentration? Please, check 9% , is it a typos?
141 64%, 65% - same comment as for line 138
Fig.1 please, explain better (in the text and in the caption) the dotted line corresponding to DMSO in the Figure 1. Please, compare the pattern of DMSO line and other lines corresponding to the tested compounds
148 60% - probably it is better if you specified here the name of the medium like you did above - BSK-II/60%
149 60% serum – please, be consistent and use the same indication for the modified medium in the entire text, e.g. BSK-II/60%
Figure 2. This figure suggests that you also tested DMSO separately, but you do not describe this in Material and Methods. Please, do it, otherwise, it is unclear. Please, include DMSO in the caption and make it clearer.
Figure 2. Please, make the colors lighter, currently all colors are very dark so that red, green and blue are almost black.
201 BSK--II revise
218 no supplementary files are provided
233 It is therefore realistic for GuaB inhibitors to play a major role here – could you please briefly explain your idea about the role of the inhibitors?
237-239 Please, insert a brief version of this phrase in the Abstract
Comments on the Quality of English Languageminor
Round 2
Reviewer 1 Report
Comments and Suggestions for Authors
The authors have provided suffecient editing to the first draft of the work which i believe is in its current state ready for publication but a minro comment is regarding answering the reviewer's comments where the answered refered lines do not match the attached manuscript and this is probably due to the track changes effect.